# PROSPECTOR: IMPROVING LLM AGENTS WITH SELF-ASKING AND TRAJECTORY RANKING

## ABSTRACT

Large language models (LLMs) have shown the ability to solve complex decision-making tasks beyond the natural language processing tasks. Current LLM agents such as ReAct can solve interactive decision-making tasks by imitating the few-shot demonstrations given in the prompt. The LLM agents based on few-shot in-context learning (ICL) achieve surprisingly high performance without training. Despite the simplicity and generalizability, the ICL-based approaches lack optimizing trajectories based on the reward from an environment. In this paper, we introduce Prospector, a reflective LLM agent that features Self-Asking and Trajectory Ranking. To elicit the LLM agent to generate more proper actions that contribute to following a given instruction, we introduce additional Self-Asking steps in the few-shot demonstrations. Furthermore, to take advantages of the stochastic generation of LLMs, we provide Trajectory Ranking in which the LLM agent generates diverse (creative) trajectories and the most rewarding trajectory is selected by using the reward prediction models. On the representative decision-making benchmark environments such as ALFWorld and WebShop, we empirically demonstrate that Prospector can considerably increase the success rate of given tasks, while outperforming recent advancements such as ReAct and Reflexion.

## 1 INTRODUCTION

Although large language models (LLMs) (Devlin et al., 2018; Radford et al., 2019; Brown et al., 2020; Raffel et al., 2020; Touvron et al., 2023) have recently shown remarkable success, it is still challenging to solve complex interactive decision-making problems that require reasoning and planning abilities (Wei et al., 2022; Nye et al., 2021). Fine-tuning the LLMs using reinforcement learning (RL) (Ouyang et al., 2022; Glaese et al., 2022; Bai et al., 2022) is one of the representative approaches to improve the reasoning and planning abilities of LLM agents. However, RL-based LLM fine-tuning methods require separate expensive training costs for each task, which are unsuitable for training LLMs with an enormous number of parameters. Recently, few-shot prompting approaches (e.g., chain-of-thought (Wei et al., 2022)) have achieved significant improvement in various natural language processing tasks (Yao et al., 2022b; Shinn et al., 2023; Wang et al., 2022), and are considered a promising direction because they do not require any fine-tuning costs of LLMs.

ReAct (Yao et al., 2022b) is one of the notable few-shot prompting approaches, which prompts LLMs to generate both verbal reasoning traces and actions in an interleaved manner. This allows the LLM agent to perform dynamic reasoning and high-level planning. However, this few-shot prompting alone may not be sufficient to generate optimal trajectories since it does not consider the task feedback signal (i.e. reward) from an environment. To leverage the task feedback signal, Shinn et al. (2023) presents Reflexion which converts the reward from the environment into verbal feedback and then uses this self-reflective feedback as additional context in the next episode. However, since Reflexion explores and reflects on task feedback signals in subsequent trials, it cannot efficiently search the diverse trajectories.

To address the aforementioned limitations, we introduce **Prospector**, a powerful LLM agent for decision-making tasks, which reinforces the ability to generate strategic actions without updating the model parameters. Prospector consists of two components *Self-Asking* and *Trajectory Ranking*. It starts with *Self-Asking* which elicits the LLM agent to generate a question and answer itself, by

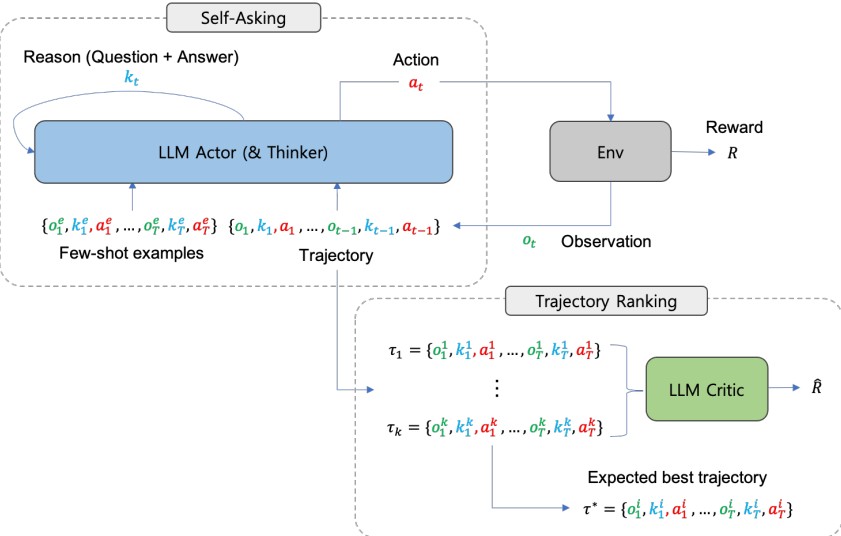

Figure 1: **Overview of Prospector.** Prospector is a reflective LLM agent that consists of two complementary LLMs such as LLM Actor and LLM Critic for solving complex interactive decision-making tasks. The LLM Actor generates actions based on the few-shot demonstrations and the history of observations and actions. To elicit the more proper actions, Prospector interleaves Self-Asking steps in the few-shot demonstrations. Furthermore, Prospector takes the advantages of the stochasticity of LLMs, and generates diverse trajectories with high temperature. Then, the LLM Critic select the most rewarding trajectory by using a reward prediction model. The LLM critic can operate either in the few-shot ICL mode or fine-tuning mode.

prompting few-shot demonstrations (i.e. in-context learning). This allows the LLM agent to collect the information necessary for decision-making on its own, and generate more strategic actions based on it. Then, to take advantage of the stochastic generation of LLMs, Prospector provides *Trajectory Ranking* in which the LLM agent generates diverse trajectories with a number of trials and then selects the most rewarding trajectory as the final action. Prospector can achieve high performance through synergizing of 1) *Self-Asking*, which can generate promising trajectory candidates, and 2) *Trajectory Ranking*, which can select the most rewarding trajectory from candidates. In the experiments, we demonstrate that Prospector outperforms recent advancements such as ReAct and Reflexion on the representative language-based interactive decision-making benchmarks including ALFWorld (Shridhar et al., 2020b) and WebShop (Yao et al., 2022a).

The contributions of this paper can be summarized as follows:

- We introduce a powerful LLM agent for decision-making tasks, which reinforces the ability to generate strategic actions without updating the model parameters (see Figure 1).
- We demonstrate that Prospector significantly outperforms ReAct (Yao et al., 2022b) and Reflexion (Shinn et al., 2023) on the representative language-based interactive decision-making benchmarks including ALFWorld (Shridhar et al., 2020b) and WebShop (Yao et al., 2022a) (see Table 2 and 5).
- We investigate the impact of the reward model used in trajectory ranking, and demonstrate that Prospector robustly generates strategic actions in both cases when using LLM-based reward model is additionally fine-tuned or not. (see Table 4 and 7).

## 2 PRELIMINARIES

### 2.1 LANGUAGE-BASED INTERACTIVE DECISION-MAKING TASKS

We consider language-based interactive decision-making tasks (Yao et al., 2022a; Shridhar et al., 2020b; Côté et al., 2019; Shridhar et al., 2020a), where reasoning and planning abilities are key

challenges in solving the task. At each time step $t$, the LLM agent receives a text observation $o_t$ from the environment, generates a text action $a_t$, and receives the associated reward $r_t$. LLM agent aims to generate the action from the context $c_t$ that maximizes the expected cumulative rewards, where $c_t := (o_1, a_1, \ldots, o_{t-1}, a_{t-1}, o_t)$. These interactive decision-making tasks can be naturally formulated as reinforcement learning (RL) problems with partially observable Markov decision processes (POMDPs) (Williams & Young, 2007). However, optimizing the LLM agent $\pi(a_t|c_t)$ with RL-based LLM fine-tuning methods requires separate expensive training costs for each task, which are unsuitable for LLMs with an enormous number of parameters.

## 2.2 FEW-SHOT PROMPTING METHODS FOR LLM AGENT

Few-shot prompting (Yao et al., 2022b; Shinn et al., 2023) is one of the representative methods that can improve the performance of LLMs without additional fine-tuning. ReAct (Yao et al., 2022b), which is the most relevant to Prospector, leverages few-shot prompting to improve reasoning skills in interactive decision-making tasks. Instead of directly generating the action from the current context $c_t$, it generates *thought* $\hat{a}_t$ corresponding to the reasoning trace based on given few-shot demonstrations including the reasoning trace, and then finally generates the action from the updated context $\hat{c}_t$ augmented with generated *thought* (i.e. $\hat{c}_t = (c_t, \hat{a}_t)$). However, reasoning the useful information directly from the context is often challenging, and this few-shot prompting alone may not be sufficient to generate optimal trajectories since it does not consider the task feedback signal (i.e. reward) from an environment.

## 3 METHOD

The overview of Prospector is shown in Figure 1. Strategic decision-making is an essential ability of LLM agents to solve complex tasks. To this end, we introduce **Prospector**, a powerful LLM agent for interactive decision-making tasks, which reinforces the ability to generate strategic actions *without updating the model parameters*. Prospector mainly consists of the following two components: 1) *Self-Asking* which elicits the LLM agent to generate a question and answer itself, by prompting few-shot demonstrations (i.e. in-context learning). 2) *Trajectory Ranking* which the LLM agent generates diverse trajectories with a number of trials and then selects the most rewarding trajectory as the final action. Combining these components allows the LLM agent to trial promising actions and generate the most strategic actions. Our algorithm can be adopted for any LLM and decision-making task, given few-shot demonstrations.

## 3.1 SELF-ASKING

In order to reinforce the LLM agent without updating the model parameters, we adopt a prominent way of few-shot prompting that additionally leads to performing intermediate steps before generating the final action. We present *Self-Asking* which elicits the LLM agent to generate a question and answer itself, by prompting few-shot demonstrations (i.e. in-context learning). Unlike ReAct (Yao et al., 2022b), which performs the *reasoning* as an intermediate step, we first attempt to perform question-and-answering as an intermediate step towards more strategic decision-making. Self-Asking first performs the *asking* about necessary information for strategic decision-making and *answering* them before generating final action (Figure 2). This sophisticated process of information collecting encourages to generate more promising actions to help achieve the task. Concrete examples of Self-Asking prompts (named as **AskAct**) can be found in Table 14 and Table 17 of the Appendix.

## 3.2 TRAJECTORY RANKING

Since generating trajectory from an LLM is relatively much cheaper than training an LLM, we consider generating trajectories and selecting the best trajectory among them instead of training the LLM. To this end, we present *Trajectory Ranking* in which the LLM agent generates diverse trajectories with a number of trials and then selects the most rewarding trajectory as the final action. Thanks to Self-Asking, which allows LLM agents to generate more promising actions, Prospector can consider high-quality trajectories as candidates for final actions. However, most real-world scenarios allow the agent to interact with the environment (i.e. simulation) but not receive rewards.

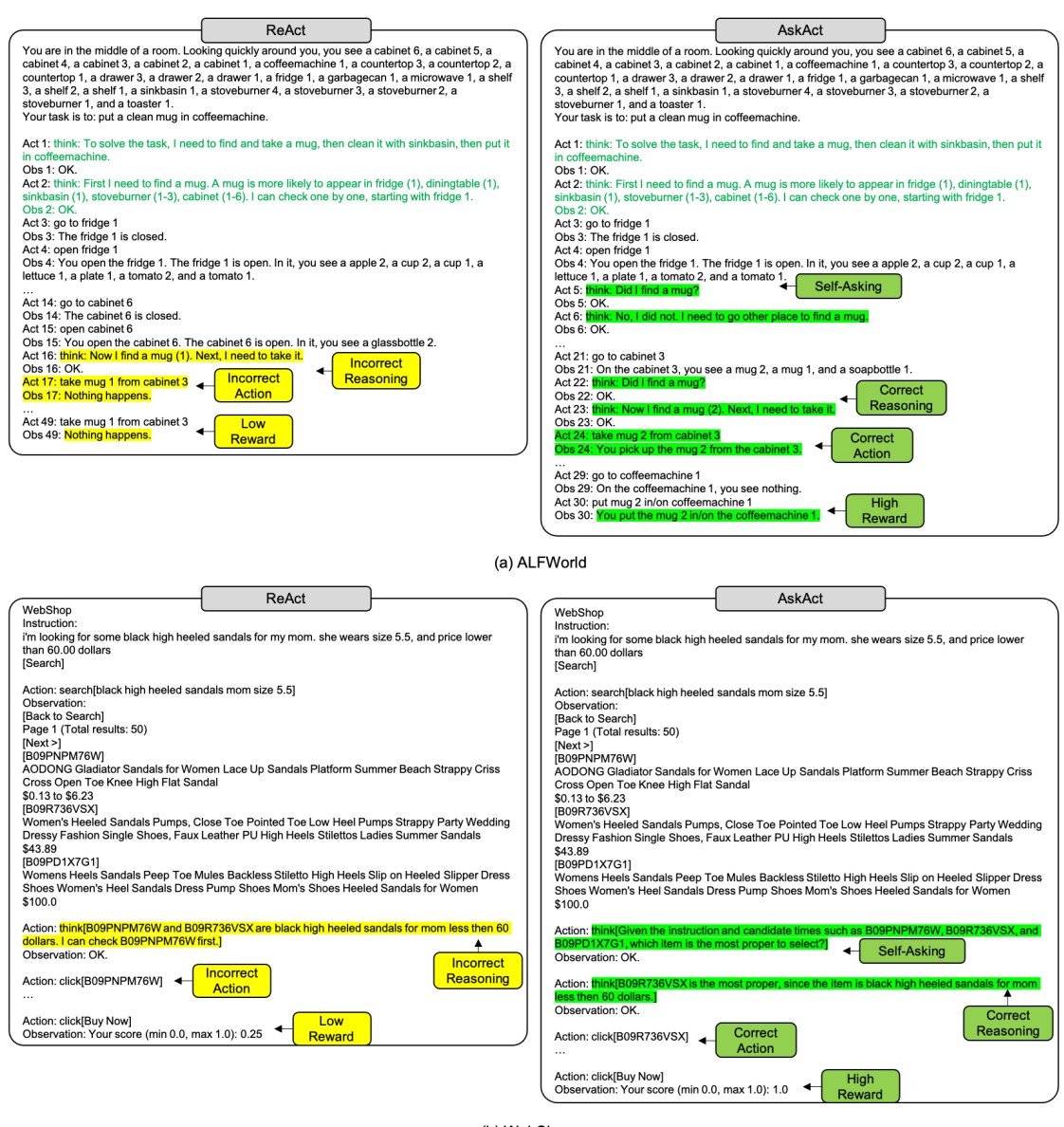

Figure 2: **Comparison of ReAct and AskAct.** AskAct is a prompting method that introduces additional self-asking steps in a ReAct prompt. (a) In ALFWorld, the self-asking step checks if a target object is found. This can elicit a correct action by alleviating hallucination. (b) In WebShop, the self-asking step explicitly tries to determine which item is the most proper. This can elicit a better item selection.

For example, in a shopping scenario such as WebShop, buyers can browse various products, but they cannot check their satisfaction (i.e. reward) by purchasing the products themselves. Therefore, we investigate two methods to estimate the trajectory reward from a given dataset: (1) Few-shot LLM critics, and (2) Fine-tuned LLM critics.

**Few-shot LLM Critics.** Motivated by recent methods of using LLMs as an evaluator (Li et al., 2023; Ye et al., 2023), we attempt to use LLMs as reward estimators for interactive decision-making tasks. To evaluate the trajectories without additional training of the reward model, we use few-shot in-context learning with reward-labeled trajectories. We provide the Critic prompt template used for

| ALFWorld | WebShop |
|---|---|
| Evaluate if the instruction given in the input is accomplished by performing a sequence of actions (fail/success). | Evaluate if the instruction given in the input is accomplished by selecting the proper item (low/middle/high). |
| ### Input:
{Example *success* trajectory}
### Response:  success | ### Input:
{Example *high-reward* trajectory}
### Response:  high |
| ### Input:
{Example *fail* trajectory}
### Response:  fail | ### Input:
{Example *low-reward* trajectory}
### Response:  low |
| ### Input:
{Input trajectory}
### Response: | ### Input:
{Input trajectory}
### Response: |

Table 1: **Critic prompt template for few-shot reward prediction.**

few-shot reward prediction in Table 1. More concrete examples of Critic prompts can be found the Table 13 and Table 15 in the Appendix.

**Fine-tuned LLM Critics.**   In some complex environments such as WebShop (Yao et al., 2022a), one of the most powerful LLMs such as GPT-3 have difficulty in reward prediction in a few-shot manner (see Table 6). In this case, open-sourced LLMs fine-tuned on trajectory data can help to increase the performance of Prospector agents. The details can be found in Table 5 and Table 7 in the Experiment section.

## 4 EXPERIMENTS

### 4.1 ALFWORLD

ALFWorld (Shridhar et al., 2020b) is a multi-modal interactive decision-making benchmark that is specialized on embodied reasoning tasks such as solving house-holding tasks. It is designed by aligning TextWorld (Côté et al., 2019), an interactive text-based game, and ALFRED (Shridhar et al., 2020a), a representative embodied AI benchmark. It includes 6 types of tasks such as (1) pick and place, (2) examine in light, (3) clean and place, (4) heat and place, (5) cool and place, and (6) pic two and place. The ALFRED dataset provides 3,553 tasks for training, 140 tasks for seen testing, and 134 tasks for unseen testing. In this paper, we perform the experiments in the text-mode of ALFWorld where a natural language instruction is given and the agent is requested to generate text-based actions by interacting the environment. We evaluate LLM agents on the unseen 134 tasks in the ALFRED dataset. For fine-tuning open-sourced LLMs for Trajectory Ranking, we use 3K training tasks in the ALFRED dataset.

### 4.1.1 SUCCESS RATE

**Comparison.**   In Table 2, we compare the success rate of Prospector with the recent LLM agents such as ReAct Yao et al. (2022b) and Reflexion (Shinn et al., 2023) on the ALFWorld environment. To show the difficulty of the tasks in the environment, we also provide the performance of BUT-LER, an agent that do not use LLMs. As shown in the table, Prospector outperforms ReAct and Reflextion. ReAct only uses few-shot demonstration for solving house-holding tasks in ALFWorld. Reflexion further improves the success rate of the ReAct agent by using the iterative refinement of LLM outputs. Here, $k = 5$ refinements are performed. For the purpose of comparison, Prospector uses ReAct as the base LLM Actor. Then, Prospector generates diverse trajectories and selects the expected best trajectory by using Trajectory Ranking (TR). Here, $k = 5$ trajectories are generated, and Trajectory Ranking selects the expected best trajectory by 2-shot LLM Critic. Prospector can further leverage AskAct prompting as the LLM Actor. The performance of LLM Critics can be found in the following sections.

| Method | LLM Actor | LLM Critic | Success Rate (%) |
|---|---|---|---|
| BUTLER | - | - | 37.0 |
| Act | PaLM-540B | - | 45.0 |
| ReAct | PaLM-540B | - | 70.9 |
| ReAct | text-davinci-002 | - | 78.4 |
| ReAct + Reflexion ($k = 5$) | text-davinci-002 | - | 86.0 |
| ReAct + TR ($k = 5$) (Prospector) | text-davinci-002 | text-davinci-002 | **95.5** |
| ReAct | Llama-2-70B | - | 41.0 |
| ReAct + TR ($k = 5$) (Prospector) | Llama-2-70B | FLAN-T5-3B (SFT) | 77.6 |
| AskAct | Llama-2-70B | - | 56.7 |
| AskAct + TR ($k = 5$) (Prospector) | Llama-2-70B | FLAN-T5-3B (SFT) | 86.6 |

Table 2: **Performance comparison of LLM agents on ALFWorld.** Prospector with Trajectory Ranking (TR) achieves better success rate than recent advancements such as ReAct Yao et al. (2022b) and Reflexion (Shinn et al., 2023) on ALFWorld.

| Method | LLM Actor | LLM Critic | $k = 1$ | 2 | 3 | 4 | 5 |
|---|---|---|---|---|---|---|---|
| ReAct + TR | Llama-2-70B | FLAN-T5-3B (SFT) | 33.6 | 59.0 | 69.4 | 73.1 | 77.6 |
| AskAct + TR | Llama-2-70B | FLAN-T5-3B (SFT) | 53.7 | 76.1 | 80.6 | 84.3 | 86.6 |
| ReAct + TR | text-davinci-002 | text-davinci-002 | 71.6 | - | 93.3 | - | **95.5** |

Table 3: **Success rate with regard to the number of trajectories.**

**Effect of the number of trials.** In Table 3, we show the change in the success rate with reward to the number of generated trajectories (trials). As shown in the figure, the success rate of Prospector increases as the number of generated trajectories ($k$) increases. To generate diverse (creative) trajectories, the LLM Actor of Prospector sets the temperature to 0.8. For Trajectory Ranking (TR), 2-shot LLM Critic (`text-davinci-002`) is used, and its temperature is set to 0.2. Since the reward prediction accuracy of 2-shot LLM Critic is very high (97% in Table 10 in the Appendix), Prospector can select the highly-rewarding trajectories from diverse trials and considerably increase the success rate.

**Task-level success rate.** We provide the detailed success rate for each task type in the ALFWorld benchmark in the Appendix section.

### 4.1.2 PERFORMANCE OF LLM CRITICS

**Few-shot accuracy of LLM Critics.** In Table 10, we show the few-shot reward prediction accuracy of LLM Critics on ALFWorld. The few-shot accuracy is high enough to be used in Trajectory Ranking without the need for find-tuning open-sourced LLMs on AFLWorld trajectory data.

| LLM Critic | Param. | Adaptation Method | # Trainable Param. | Accuracy (success/fail) |
|---|---|---|---|---|
| text-davinci-003 | - | 2-shot ICL | 0 | 95.5 |
| text-davinci-002 | - | 2-shot ICL | 0 | **97.0** |
| Bloom | 7.1B | LoRA FT on 3K data | 3.9M | 79.1 |
| Llama-2-Chat | 7B | LoRA FT on 3K data | 4.2M | 94.8 |
| Bloomz | 7.1B | LoRA FT on 3K data | 3.9M | 95.5 |
| Llama-2 | 7B | LoRA FT on 3K data | 4.2M | 96.3 |
| GPT-J | 6B | LoRA FT on 3K data | 3.7M | 97.3 |
| T5 | 3B | LoRA FT on 3K data | 5.9M | 98.5 |
| FLAN-T5 | 3B | LoRA FT on 3K data | 4.7M | **98.5** |

Table 4: **Fine-tuning reward prediction accuracy of LLM Critics on ALFWorld.**

| Method | LLM Actor | LLM Critic | Reward | Success Rate |
|---|---|---|---|---|
| Human (expert) | - | - | 82.1 | 59.6 |
| Human (average) | - | - | 75.5 | 50.0 |
| IL | - | - | 59.9 | 29.1 |
| IL + RL | - | - | 62.4 | 28.7 |
| ReAct | text-davinci-002 | - | 63.3 | 35.8 |
| ReAct + Reflexion ($k = 8$) | text-davinci-002 | - | - | 35.0 |
| AskAct (Prospector) | text-davinci-002 | - | 66.5 | 39.8 |
| AskAct + TR ($k = 8$) (Prospector) | text-davinci-002 | text-davinci-002 | 69.3 | 41.4 |
| AskAct + TR ($k = 8$) (Prospector) | text-davinci-002 | Llama-2-7B-Chat (SFT) | **70.8** | 43.0 |
| AskAct + TR ($k = 8$) (Prospector) | text-davinci-002 | Oracle (w/ reward) | 71.3 | 47.0 |
| ReAct | Llama-2-70B | - | 62.3 | 37.6 |
| AskAct (Prospector) | Llama-2-70B | - | 68.6 | 42.2 |
| ReAct + TR ($k = 8$) (Prospector) | Llama-2-70B | FLAN-T5-3B (SFT) | 69.3 | 42.2 |
| AskAct + TR ($k = 8$) (Prospector) | Llama-2-70B | FLAN-T5-3B (SFT) | 70.2 | **43.6** |

Table 5: **Performance comparison of LLM agents on WebShop.** Prospector with Self-Asking (AskAct) and Trajectory Ranking (TR) achieves better success rate than the recent advancements such as ReAct (Yao et al., 2022b), and Reflexion (Shinn et al., 2023).

**Fine-tuning accuracy of LLM Critic.** In Table 4, we show the fine-tuning reward prediction accuracy of LLM Critics on ALFWorld. We finetune open-sourced LLMs on 3K ALFWorld trajectory data. For decoder-only models, we choose GPT-J (Wang, 2021), Bloom (Scao et al., 2022), Bloomz (Muennighoff et al., 2022), and Llama-2 (Touvron et al., 2023). For encoder-decoder models, we choose T5 (Raffel et al., 2020) and FLAN-T5 (Chung et al., 2022). For parameter-efficient fine-tuning, we use LoRA (Hu et al., 2021). By fine-tuning open-sourced LLMs on 3K ALFWorld trajectory data, they can achieve comparable or better reward prediction accuracy with the closed LLMs such as `text-davinci-002`. The hyperparameters used for fine-tuning LLM Critics can be found in Table 18 in the Appendix.

## 4.2 WEBSHOP

WebShop (Yao et al., 2022a) is a large-scale online shopping environment with more than 1M real-world products crawled from Amazon. The agent is given a natural language instruction (e.g., "I would like 3 ounce bottle of bright citrus deodorant for sensitive skin, and price lower than 50.00 dollars."), and required to make a sequence of actions (e.g., querying the search engine with keywords and clicking on a product title) to accomplish the given instruction. More specifically, the task mainly consists of five stages: (1) searching products with query words, (2) selecting a product in the search results, (3) selecting proper options, (4) reviewing the product details, and (5) clicking on the "Buy Now" button. WebShop provides two modes: (1) multi-modal mode with product figures, and (2) text-based mode. Also, WebShop provides about 12K human instructions, and reserves 500 instructions for tasting. In this paper, we perform experiments in the text-based mode and evaluate LLM agents on the official 500 test instructions. We use 12K human instructions (without test instruction) for generating trajectories and fine-tuning LLM Critics on them.

### 4.2.1 SUCCESS RATE

In Table 5, we compare the performance of Prospector with the recent LLM agents such as Re-Act (Yao et al., 2022b) and Reflexion (Shinn et al., 2023). In addition to these methods, to assess the difficulty of the environment, we also provide the performance of human as upper bound, and the performance of the traditional methods that use Imitation Learning (IL) or Reinforcement Learning (RL) as strong baselines. These results are quoted from the WebShop (Yao et al., 2022a) paper. As shown in the table, Prospector achieves better success rate (43.0%) than the recent advancements such as ReAct (35.8%) and Reflexion(35.0%). Compared to the traditional IL (29.1%) and RL (28.7%) methods, ReAct agents based on `text-davinci-002` surprisingly achieve high success rate without training. However, there is a gap with the human performance (50%). Prospector can considerably reduce this gap by using Self-Asking and Trajectory Ranking (TR). Prospector using

| LLM Critic | 1-shot | 2-shot | 3-shot |
|---|---|---|---|
| text-davinci-002 | 34.4 | **47.0** | 42.4 |
| text-davinci-003 | 37.0 | 42.2 | 36.2 |

Table 6: **Few-shot reward prediction accuracy of LLM Critics on WebShop.** Few-shot LLM Critics have some difficulty in predicting the reward of the agent's trajectory in a complex environment such as WebShop. This requires LLM Critics fine-tuned on WebShop trajectory data.

| LLM Critic | Param. | Adaptation Method | # Trainable Param. | Accuracy (hi/mi/lo) |
|---|---|---|---|---|
| text-davinci-003 | - | 2-shot ICL | 0 | 42.2 |
| text-davinci-002 | - | 2-shot ICL | 0 | 47.0 |
| Bloom | 7.1B | LoRA FT on 12K data | 3.9M | 67.2 |
| GPT-J | 6B | LoRA FT on 12K data | 3.7M | 72.0 |
| Llama-2 | 7.1B | LoRA FT on 12K data | 4.2M | 73.8 |
| Bloomz | 7.1B | LoRA FT on 12K data | 3.9M | 75.8 |
| Llama-2-Chat | 7B | LoRA FT on 12K data | 4.2M | 76.2 |
| T5 | 3B | LoRA FT on 12K data | 5.9M | 77.0 |
| FLAN-T5 | 3B | LoRA FT on 12K data | 4.7M | **78.0** |

Table 7: **Fine-tuning reward prediction accuracy of LLM Critics on WebShop.** Fine-tuned LLM Critics (e.g., Llama-2 fine-tuned on 12K trajectory data) provide significantly improved reward prediction accuracy, compared to few-shot LLM Critics (e.g., `text-davinci-002`) in WebShop. Improved prediction accuracy of fine-tuned LLM Critics help to increase the success rate of Prospector

AskAct prompting can increase the success rate up to 39.8% compared to ReAct (35.8%). Prospector with AskAct and TR further increase the success rate up to 41.3%. However, since the few-shot LLM Critic based on `text-davinci-002` does not provide high accuracy (47.0%) in reward prediction, the improvement is not significant. In contrast, since the fine-tuned LLM Critic based on Llama-2-7B-Chat (Touvron et al., 2023) provides much higher accuracy in reward prediction, Prospector can achieve better success rate (43.0%). Note that if the oracle with known reward is used, the success rate can be reached by up to 47.0%, while considerably closing the gap with the human performance (50%). Note that the performance of LLM Critic is important to improve the performance of LLM agents. Regarding this, we provide the detailed additional experiments on LLM Critics in Table 6, Table 7, and Table 8 in the following subsection.

### 4.2.2 Performance of LLM Critics

**Few-shot accuracy of LLM Critics.** In Table 6, we provide the few-shot reward prediction accuracy of API-based LLM Critics such as `text-davinci-002` on WebShop. We find that few-shot LLM Critics have some difficulty in predicting the reward of a given trajectory in a complex environment such as WebShop. LLM Critics with low reward prediction accuracy can not be used for reliable Trajectory Ranking (TR). This result requires us to fine-tune open-sourced LLMs such as Llama-2 on WebShop trajectory data.

**Fine-tuning accuracy of LLM Critics.** In Table 7, we compare the reward prediction accuracy of fine-tuned LLM Critics. We finetune open-sourced LLMs on 3K ALFWorld trajectory data. For decoder-only models, we choose GPT-J (Wang, 2021), Bloom (Scao et al., 2022), Bloomz (Muennighoff et al., 2022), and Llama-2 (Touvron et al., 2023). For encoder-decoder models, we choose T5 (Raffel et al., 2020) and FLAN-T5 (Chung et al., 2022). For parameter-efficient fine-tuning, we

| LLM Critic | 3K | 6K | 9K | 12K |
|---|---|---|---|---|
| Llama-2-7B-Chat (LoRA) | 70.0 | 71.1 | 76.2 | **76.2** |

Table 8: **Fine-tuning accuracy over the dataset size.**

use LoRA (Hu et al., 2021). Fine-tuned LLM Critics (e.g., Llama-2 fine-tuned on 12K trajectory data) provide significantly improved reward prediction accuracy, compared to few-shot LLM Critics (e.g., `text-davinci-002`) in the WebShop environment. Improved prediction accuracy of fine-tuned LLM Critics help to increase the success rate of Prospector.

In Table 8, we provide the change in reward prediction accuracy with regard to the size of trajectory data. We can see that the reward prediction accuracy increases as the data size increases. The hyperparameters used for fine-tuning LLM Critics can be found in Table 18 in the Appendix.

## 5 RELATED WORK

**Reasoning in LLMs.** Few-shot in-context learning (ICL) is one of the representative methods, that achieves high performance in various NLP tasks. However, ICL-based approaches are known to struggle in reasoning tasks. To address this shortcoming, Wei et al. (2022) introduced chain-of-thoughts (CoT) that generates a series of short sentences that mimic the human reasoning process. CoT with Self-Consistency (CoT-SC) (Wang et al., 2022) samples $k$ diverse reasoning paths instead of selecting the greedy one and subsequently returns the most frequent answer. However, since this approach is only applicable when the output space is limited, Tree-of-Thoughts (ToT) (Yao et al., 2023) overcomes this limitation by generalizing CoT prompting and further enhancing local exploration of thought. On the other hand, Self-Ask (Press et al., 2022) improves CoT on QA tasks by transforming a chain-of-thought into a multi-turn self-question-answering process. This study also introduced the concept of conducting reasoning through question-answering concurrently with our work, but we want to emphasize that while Self-Ask focuses on QA tasks, our work enhances LLM agents for interactive decision-making tasks through synergizing self-asking and trajectory ranking.

**LLM Agents.** The use of reasoning prompts for LLM agents also enables achieving high performance in text-based interactive decision-making tasks without training. ReAct (Yao et al., 2022b) is an algorithm that integrates reasoning and action within language models to tackle a diverse range of language reasoning and decision-making tasks. When task feedback is accessible, Reflexion (Shinn et al., 2023) and Self-Refine (Madaan et al., 2023) reinforce LLM agents by learning a linguistic reward model to verbally reflect the task feedback signal. We note that Reflexion iteratively generates trajectories and reflects rewards verbally in sequence, while Prospector generates diverse trajectories in parallel and chooses the best one in terms of rewards. On the other hand, when human or other external knowledge sources are available, Asking-Before-Action (ABA) (Chen et al., 2023) incorporates humans into the decision-making process by introducing a contextual MDP with human or external information sources in the loop.

**Reward Models and Rankings.** Reward models and rankings are widely employed within the LLM context and their applications. In order to enhance LLM performance, InstructGPT (Ouyang et al., 2022) and Llama-2 (Touvron et al., 2023) leverage RL for fine-tuning the LLMs themselves. Furthermore, LLMs have showcased their impressive capability to generate code across diverse programming tasks, highlighting their versatility. Within this domain, a neural ranker, CodeRanker (Inala et al., 2022), was introduced to improve the accuracy of various code generation models, enabling them to predict the correctness of sampled code without actual execution. On the other hands, to harness the LLM's semantic knowledge about the real world, SayCan (Brohan et al., 2023) proposed an innovative approach to combine LLM and RL.

## 6 LIMITATIONS AND CONCLUSION

**Limitations.** ReAct is more efficient than Prospector. However, Prospector performs much better. Also, Trajectory Ranking is further more efficient and performs better than Reflexion.

**Conclusion.** In this paper, we introduce Prospector, an improved LLM agent that features Trajectory Ranking and Self-Asking. Self-Asking steps interleaved in the few-shot demonstration elicit the LLM agent to generate more proper actions to follow the given instructions. Trajectory Ranking enables the LLM agent to generate diverse trajectories and select the most rewarding trajectory by using reward prediction models.

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

# 7 APPENDIX

## 7.1 ADDITIONAL EXPERIMENTS

### 7.1.1 ALFWORLD

**Task-level success rate.** In Table 9, we provide the detailed success rate for each task type in the ALFWorld benchmark.

| Method | LLM Actor | Pick | Clean | Heat | Cool | Look | Pick2 | All (%) |
|---|---|---|---|---|---|---|---|---|
| BUTLER | - | 46 | 39 | 74 | 100 | 22 | 24 | 37 |
| Act | PaLM-540B | 88 | 42 | 74 | 67 | 72 | 41 | 45 |
| ReAct | PaLM-540M | 92 | 58 | **96** | 86 | 78 | 41 | 71 |
| ReAct | text-davinci-002 | 88 | 61 | 78 | 86 | 89 | 71 | 78 |
| | (#success/#tasks) | 21/24 | 19/31 | 18/23 | 18/21 | 16/18 | 12/17 | 104/134 |
| **Prospector** | text-davinci-002 | **100** | **97** | 96 | **90** | **94** | **100** | **96** |
| (w/ TR ($k = 10$)) | (#success/#tasks) | 24/24 | 30/31 | 22/23 | 19/21 | 17/18 | 17/17 | 129/134 |

Table 9: **Comparison of task-level success rate on ALFWorld.**

**Few-shot accuracy of LLM Critics.** In Table 10, we provid few-shot accuracy of LLM Critics on ALFWorld.

| LLM Critic | 1-shot | 2-shot | 3-shot |
|---|---|---|---|
| text-davinci-002 | 94.8 | **97.0** | 95.5 |
| text-davinci-003 | 93.3 | 95.5 | 94.0 |

Table 10: **Few-shot reward prediction accuracy of LLM Critics on ALFWorld.**

## 7.2 PROMPTS

### 7.2.1 ALFWORLD

I this section, we provide the prompts used for the LLM Actor and few-shot LLM Critic to solve the tasks in the ALFWorld. The ReAct prompt for the LLM Actor is provided in Table 12. The Critic prompt for the LLM Critic is provided in Table 13.

### 7.2.2 WEBSHOP

I this section, we provide the prompts used for the LLM Actor and few-shot LLM Critic to solve the decision-making tasks in the WebShop. The AskAct prompt for the LLM Actor is provided in Table 14. The Critic prompt for the few-shot LLM Critic is provided in Table 15.

## 7.3 TRAJECTORIES

### 7.3.1 ALFWORLD

In Table 16, we provide an example success trajectory generated by Prospector on ALFWorld.

### 7.3.2 WEBSHOP

In Table 17, we provide an example high-reward trajectory generated by Prosepctor on WebShop.

## 7.4 HYPER-PARAMETERS

In Table 18, we provide the hyper-parameters used for fine-tuning the LLM Critic on the trajectory data.

ReAct Prompt for ALFWorld

```
You are in the middle of a room.  Looking quickly around you,
you see a cabinet 4, a cabinet 3, a cabinet 2, a cabinet 1,
a countertop 1, a garbagecan 1, a handtowelholder 2,
a handtowelholder 1, a sinkbasin 2, a sinkbasin 1, a toilet 1,
a toiletpaperhanger 1, and a towelholder 1.

Your task is to:  put some spraybottle on toilet.

> think:  To solve the task, I need to find and take a sparybottle,
then put it on toilet.
OK.
> think:  First I need to find a spraybottle.  A spraybottle is
more likely to appear in cabinet (1-4), countertop (1), toilet (1),
sinkbasin (1-2), garbagecan (1).  I can check one by one,
starting with cabinet 1.
OK.
> go to cabinet 1
On the cabinet 1, you see a cloth 1, a soapbar 1, a soapbottle 1.
> go to cabinet 2
The cabinet 2 is closed.
> open cabinet 2
You open the cabinet 2.  The cabinet 2 is open.
In it, you see a candle 1, and a spraybottle 2.
> think:  Now I find a spraybottle (2).  Next, I need to take it.
OK.
> take spraybottle 2 from cabinet 2
You pick up the spraybottle 2 from the cabinet 2.
> think:  Now I take a spraybottle (2).  Next, I need to
put it in/on toilet 1.
OK.
> go to toilet 1
On the toilet 1, you see a soapbottle 2.
> put spraybottle 2 in/on toilet 1
You put the spraybottle 2 in/on the toilet 1.
```

Table 11: **ReAct prompt for the LLM Actor on ALFWorld.**

AskAct Prompt for ALFWorld

```
You are in the middle of a room.  Looking quickly around you,
you see a cabinet 4, a cabinet 3, a cabinet 2, a cabinet 1,
a countertop 1, a garbagecan 1, a handtowelholder 2,
a handtowelholder 1, a sinkbasin 2, a sinkbasin 1, a toilet 1,
a toiletpaperhanger 1, and a towelholder 1.

Your task is to:  put some spraybottle on toilet.

> think:  To solve the task, I need to find and take a sparybottle,
then put it on toilet.
OK.
> think:  First I need to find a spraybottle.  A spraybottle is
more likely to appear in cabinet (1-4), countertop (1), toilet (1),
sinkbasin (1-2), garbagecan (1).  I can check one by one,
starting with cabinet 1.
OK.
> go to cabinet 1
On the cabinet 1, you see a cloth 1, a soapbar 1, a soapbottle 1.
> think:  Did I find a spraybottle?
OK.
> think:  No, I did not.  I need to go other place to find a spraybottle.
OK.
> go to cabinet 2
The cabinet 2 is closed.
> open cabinet 2
You open the cabinet 2.  The cabinet 2 is open.  In it,
you see a candle 1, and a spraybottle 2.
> think:  Did I find a spraybottle?
OK.
> think:  Now I find a spraybottle (2).  Next, I need to take it.
OK.
> take spraybottle 2 from cabinet 2
You pick up the spraybottle 2 from the cabinet 2.
> think:  Now I take a spraybottle (2).  Next, I need to
put it in/on toilet 1.
OK.
> go to toilet 1
On the toilet 1, you see a soapbottle 2.
> put spraybottle 2 in/on toilet 1
You put the spraybottle 2 in/on the toilet 1.
```

Table 12: **AskAct prompt for the LLM Actor on ALFWorld.**

---

Critic Prompt on ALFWorld

---

Evaluate if the instruction given in the input is accomplished by
performing a sequence of actions (fail/success).

### Input:
You are in the middle of a room.  Looking quickly around you,
you see a cabinet 2, a cabinet 1, a coffeemachine 1, a countertop 3,
a countertop 2, a countertop 1, a diningtable 1, a drawer 6,
a drawer 5, a drawer 4, a drawer 3, a drawer 2, a drawer 1,
a fridge 1, a garbagecan 1, a microwave 1, a sinkbasin 1,
a stoveburner 4, a stoveburner 3, a stoveburner 2, a stoveburner 1,
and a toaster 1.

Your task is to:  clean some lettuce and put it in fridge.

> think:  To solve the task, I need to find and take a lettuce,
then clean it with sinkbasin, then put it in fridge.
OK.
> think:  First I need to find a lettuce.  A lettuce is more likely
to appear in fridge (1), diningtable (1), sinkbasin (1),
stoveburner (1-3), cabinet (1-2).  I can check one by one,
starting with fridge 1.
OK.
> go to fridge 1
The fridge 1 is closed.
> open fridge 1
You open the fridge 1.  The fridge 1 is open.  In it, you see a apple 1,
a cup 1, a lettuce 1, a potato 2, and a potato 1.
> think:  Now I find a lettuce (1).  Next, I need to take it.
OK.
> take lettuce 1 from fridge 1
You pick up the lettuce 1 from the fridge 1.
> think:  Now I take a lettuce (1).  Next, I need to go to
sinkbasin (1) and clean it.
OK.
> go to sinkbasin 1
On the sinkbasin 1, you see a apple 2, a butterknife 1, a cup 2,
a knife 2, a potato 3, and a spoon 3.
> clean lettuce 1 with sinkbasin 1
You clean the lettuce 1 using the sinkbasin 1.
> think:  Now I clean a lettuce (1).  Next, I need to put it
in/on fridge 1.
OK.
> go to fridge 1
The fridge 1 is open.  In it, you see a apple 1, a cup 1, a potato 2,
and a potato 1.
> put lettuce 1 in/on fridge 1
You put the lettuce 1 in/on the fridge 1.
### Response:  success

### Input:
{Input trajectory}
### Response:

---

Table 13: **Critic prompt for the few-shot LLM Critic on ALFWorld.** The *success* trajectory given as a 1-shot demonstration is generated by `text-davinci-002` with ReAct prompting.

---

AskAct Prompt for WebShop

---

```
Webshop
Instruction:
i would like a 3 ounce bottle of bright citrus deodorant for
sensitive skin, and price lower than 50.00 dollars
[Search]

Action:  search[3 ounce bright citrus deodorant sensitive skin]
Observation:
[Back to Search]
Page 1 (Total results:  50)
[Next >]
[B078GWRC1J]
Bright Citrus Deodorant by Earth Mama | Natural and Safe for
Sensitive Skin, Pregnancy and Breastfeeding,
Contains Organic Calendula 3-Ounce
$10.99
[B078GTKVXY]
Ginger Fresh Deodorant by Earth Mama | Natural and Safe for
Sensitive Skin, Pregnancy and Breastfeeding,
Contains Organic Calendula 3-Ounce
$10.99
[B08KBVJ4XN]
Barrel and Oak - Aluminum-Free Deodorant, Deodorant for Men,
Essential Oil-Based Scent, 24-Hour Odor Protection,
Cedar & Patchouli Blend, Gentle on Sensitive Skin
(Mountain Sage, 2.7 oz, 2-Pack)
$15.95

Action:  think[Given the instruction and candidate times
such as B078GWRC1J, B078GTKVXY, and B08KBVJ4XN, which item is
the most proper to select?]
Observation:  OK.
Action:  think[B078GWRC1J is the most proper, since the item
is bright citrus deodorant less then 50 dollars.]
Observation:  OK.

Action:  click[B078GWRC1J]
Observation:
[Back to Search]
[< Prev]
scent [assorted scents][bright citrus][calming lavender]
[ginger fresh][simply non-scents]
size [travel set (4-pack)][3 ounce (pack of 1)][3-ounce (2-pack)]
Bright Citrus Deodorant by Earth Mama | Natural and Safe for
Sensitive Skin, Pregnancy and Breastfeeding,
Contains Organic Calendula 3-Ounce
Price:  $10.99
Rating:  N.A.
[Description]
[Features]
[Reviews]
[Buy Now]

Action:  think[For 3 ounce bottle of bright citrus deodorant for
sensitive skin, the item has options 'bright citrus' and
'3 ounce (pack of 1)' and seems good to buy.]
Observation:  OK.

Action:  click[bright citrus]
Observation:  You have clicked bright citrus.
Action:  click[3 ounce (pack of 1)]
Observation:  You have clicked 3 ounce (pack of 1).
Action:  click[Buy Now]
```

---

Table 14: **AskAct prompt for the LLM Actor on WebShop.**

```
Critic Prompt on WebShop

Evaluate if the instruction given in the input is accomplished
by selecting the proper item (low/middle/high).

### Input:
WebShop
Instruction:
find me a regular fit machine washable cargo pants with buttoned closure
in 6057 apricot color and 29 size, and price lower than 40.00 dollars
[Search]

Action:  search[regular fit machine washable cargo pants buttoned closure
6057 apricot color 29 size]
Observation:
[Back to Search]
Page 1 (Total results:  50)
[Next >]
[B00PQFK67G]
Match Men's Athletic-Fit Cargo Pants
$21.99 to $29.99
[B01M4KXXPT]
Vertx Men's Phantom Lt 2.0 Shorts
$21.14 to $76.84
[B01EAP5JD8]
DUNLOP Mens On Site Workwear Trousers Regular Classic
$63.99

Action:  think[B00PQFK67G is the one I want.]
Observation:  OK.

Action:  click[B00PQFK67G]
Observation:
[Back to Search]
[< Prev]
size [29][30][32][34][36][38][40][42][44]
color [6057 apricot][6057 khaki][6062 dark khaki][6531 apricot][6531 black]
 [6531 grayish black][6531 light gray][6531 light khaki][khaki max]
 [6057 light apricot][6069 blue][6531 army green][6531 blue][6531 dark gray]
 [6071 black][6071 dark blue][6071 light khaki][6039m tangerine max]
 [6071 dark apricot][6071 dark khaki][6057 bluish purple][6057 dark gray]
 [6071 brown][6071 gray][6039m multi-max][6039 light khaki][6071 steel gray]
 [6039 dark yellow][6039 blue][6039 apricot][6039 dark gray][6050 peacock blue]
 [6057 silver gray][6066 light apricot]
Match Men's Athletic-Fit Cargo Pants
Price:  $21.99 to $29.99
Rating:  N.A.
[Description]
[Features]
[Reviews]
[Attributes]
[Buy Now]

Action:  think[The item has options '29', '6057 apricot', '6057 khaki',
'6062 dark khaki', '6531 apricot', '6531 black', '6531 grayish black',
'6531 light gray', '6531 light khaki', 'khaki max', '6057 light apricot',
'6069 blue', '6531 army green', '6531 blue', '6531 dark gray', '6071 black']
Observation:  OK.

Action:  click[29]
Observation:  You have clicked 29.

Action:  click[6057 apricot]
Observation:  You have clicked 6057 apricot.

Action:  click[Buy Now]
### Response:  high

### Input:
{Input trajectory}
### Response:
```

Table 15: **Critic prompt for the few-shot LLM Critic on WebShop.** The *high-reward* trajectory given as a 1-shot demonstration is generated by `text-davinci-002` with ReAct prompting.

Success Trajectory on ALFWorld

```
You are in the middle of a room. Looking quickly around you,
you see a cabinet 6, a cabinet 5, a cabinet 4, a cabinet 3, a cabinet 2,
a cabinet 1, a coffeemachine 1, a countertop 3, a countertop 2,
a countertop 1, a drawer 3, a drawer 2, a drawer 1, a fridge 1,
a garbagecan 1, a microwave 1, a shelf 3, a shelf 2, a shelf 1,
a sinkbasin 1, a stoveburner 4, a stoveburner 3, a stoveburner 2,
a stoveburner 1, and a toaster 1.

Your task is to:  clean some bowl and put it in cabinet.
> think:  To solve the task, I need to find and take a bowl,
then clean it with sinkbasin, then put it in cabinet.
OK.
> think:  First I need to find a bowl.  A bowl is more likely
to appear in countertop (1-3), drawer (1-3), shelf (1-3), garbagecan (1),
sinkbasin (1).  I can check one by one, starting with countertop 1.
OK.
> go to countertop 1
On the countertop 1, you see a apple 3, a egg 2, and a vase 2.
> go to countertop 2
On the countertop 2, you see a creditcard 2, a creditcard 1,
a cup 3, a cup 2, a fork 2, a glassbottle 1, a knife 2, a knife 1,
a lettuce 2, a lettuce 1, and a pan 2.
> go to countertop 3
On the countertop 3, you see a bread 1, a butterknife 1,
a cellphone 1, a houseplant 1, a papertowelroll 1, a plate 1,
a soapbottle 1, a spoon 3, a spoon 2, a statue 1, and a vase 3.
> go to drawer 1
The drawer 1 is closed.
> open drawer 1
You open the drawer 1.  The drawer 1 is open.
In it, you see a fork 1.
> go to drawer 2
The drawer 2 is closed.
> open drawer 2
You open the drawer 2.  The drawer 2 is open.
In it, you see a fork 3, a spatula 2, a spatula 1, and a spoon 1.
> go to drawer 3
The drawer 3 is closed.
> open drawer 3
You open the drawer 3.  The drawer 3 is open.
In it, you see a dishsponge 1.
> go to shelf 1
On the shelf 1, you see nothing.
> go to shelf 2
On the shelf 2, you see a bowl 2, and a peppershaker 1.
> think:  Now I find a bowl (2).  Next, I need to take it.
OK.
> take bowl 2 from shelf 2
You pick up the bowl 2 from the shelf 2.
> think:  Now I take a bowl (2).  Next, I need to go
to sinkbasin (1) and clean it.
OK.
> go to sinkbasin 1
On the sinkbasin 1, you see nothing.
> clean bowl 2 with sinkbasin 1
You clean the bowl 2 using the sinkbasin 1.
> think:  Now I clean a bowl (2).  Next, I need to put it
in/on cabinet 1.
OK.
> go to cabinet 1
On the cabinet 1, you see nothing.
> put bowl 2 in/on cabinet 1
You put the bowl 2 in/on the cabinet 1.
```

Table 16: **Success trajectory generated by Prospector on ALFWorld.**

---

High-reward trajectory on WebShop

---

WebShop
```
Instruction:
i am looking for mn4 color foundation for my sensitive skin,
and price lower than 40.00 dollars
[Search]

Action:  search[mn4 color foundation sensitive skin]
Observation:
[Back to Search]
Page 1 (Total results:  50)
[Next >]
[B077PR9TL4]
Physicians Formula The Healthy Foundation with SPF 20, LW2, 1 Fl Oz
$7.59
[B07VLKYJNH]
PÜR 4-in-1 Love Your Selfie Longwear Foundation & Concealer - Full Coverage
Foundation & Concealer Makeup For Dark Spots, Blemishes and Imperfections -
Long-Lasting Liquid Makeup & Skincare
$36.0
[B09J1X9M1P]
Neutrogena Healthy Skin Sensitive Skin Serum Foundation with Pro-Vitamin B5,
Color Correcting & Pore Minimizing Liquid Foundation & Face Serum, Buildable
Coverage, Medium 02, 1 oz
$14.39
```
```
Action:  think[Given the instruction and candidate times such as B077PR9TL4,
B07VLKYJNH, and B09J1X9M1P, which item is the most proper to select?]
Observation:  OK.
```
```
Action:  think[B077PR9TL4 is the most proper, since the item is mn4 color
foundation less then 40 dollars.]
Observation:  OK.
```
```
Action:  click[B077PR9TL4]
Observation:
[Back to Search]
[< Prev]
color [1][dc1][dn3][dn4][lc1][ln3][ln4][lw2][mc1][mn3][mn4][mw2]
size [1 fl oz (pack of 1)][1 ounce (pack of 1)][1 fluid ounce]
Physicians Formula The Healthy Foundation with SPF 20, LW2, 1 Fl Oz
Price:  $7.59
Rating:  N.A.
[Description]
[Features]
[Reviews]
[Attributes]
[Buy Now]
```
```
Action:  think[For mn4 color foundation for sensitive skin, the item has
options 'mn4' and '1 fl oz (pack of 1)' and seems good to buy.]
Observation:  OK.
```
```
Action:  click[mn4]
Observation:  You have clicked mn4.
```
```
Action:  click[1 fl oz (pack of 1)]
Observation:  You have clicked 1 fl oz (pack of 1).
```
```
Action:  click[Buy Now]
Observation:  Your score (min 0.0, max 1.0):  1.0
```

Table 17: **High-reward trajectory generated by Prospector on WebShop.**

| Hyper-parameter | Value |
| --- | --- |
| model max length (context length) | 1024 |
| batch size | 128 |
| max epochs | 50 |
| optimizer | AdamW |
| learning rate | 3e-4 |
| weight decay | 0.1 |
| learning rate scheduler | cosine |
| warm-up steps | 50% of the max steps |
| LoRA r | 8 |
| LoRA alpha | 32 |
| LoRA drop-out | 0.1 |

Table 18: **Hyper-parameters for LLM Critic fine-tuning.**

