# OpenReview forum: "Prospector: Improving LLM Agents with Self-Asking and Trajectory Ranking"
_ICLR.cc/2024/Conference — Submitted to ICLR 2024_

### Official Review · Reviewer_EvHL · 2023-10-30

**Soundness:** 2 fair
**Presentation:** 3 good
**Contribution:** 2 fair
**Rating:** 5
**Confidence:** 4

**Summary:**

This paper proposes Prospector, a large language model-based agent that aims to solve multi-step decision making tasks. Prospector comprises two parts: Self-Asking, which enables the LLM agent to generate questions and answer them itself, leading to improved intermediate steps in the decision making process, and Trajectory Ranking, where an LLM critic is used to predict the rewards of different trajectories. Compared to prior LLM agents for the ALFWorld and Webshop tasks that only use ICL methods, Prospector is able to achieve higher success rates.

**Strengths:**

- The proposed method is both simple and intuitive, using LLMs for both planning and critiquing of possible trajectories. The paper is well-written and clear.
- The experiments seem thorough, with comparisons against state-of-the-art methods in the same task domains, and ablations of each component of the proposed Prospector method (removing the trajectory ranking, evaluating the accuracy of the different LLM critics, comparing few-shot and finetuned LLM critics). In particular, studying the choice of either a fine-tuned or ICL-based critic is interesting and seems novel.

**Weaknesses:**

While the method is straightforward and intuitive with impressive experimental results, my main concern is that the two main components of the methods seem to lack novelty in themselves. This can maybe be clarified with further experimentation:

-  It’s not clear how much of the overall performance improvement is just due to giving the LLM multiple attempts at a single question with the trajectory ranking process. Further experiments disentangling this would be helpful: for example, if we used the same LLM critics and trajectory ranking process with the ReAct prompt, would it perform on par with Prospector (these experiments seem to be present for ALFWorld but not WebShop)? Would majority voting at every step, which also allows multiple trajectory attempts but without an explicit LLM critic, be less useful than using the LLM-based critic as in Prospector?

- The AskAct process is does not seem like a novel contribution in and of itself, as it was proposed in Measuring and Narrowing the Compositionality Gap in Language Models (Press et al., 2022). While the authors note that that Self-Ask work was developed for QA tasks specifically, applying the same general technique of prompting the LLM to ask itself a limited set of questions to reason is a limited contribution. In particular, it seems like AskAct was not applied to the ALFWorld benchmark for the Prospector agent, and only tested in WebShop as a single fixed question that asks "which observed object is most proper to select" (shown in Figure 2 and Table 13 and 16). Further experiments eliciting different types of self-asked questions across all the tasks would strengthen this contribution.

**Questions:**

- It would be helpful to have an ablation with the trajectory ranking only, and no intermediate self-asking step for WebShop, as is shown in ALFWorld. Is most of the juice coming the critic-based trajectory ranking?
- It would also be interesting to see if the few-shot critic performance improves with intermediate chain-of-thought reasoning steps in the critic prompt, rather than just having the critic immediately output a response as success/failure.
- It would help to add a clarifying caption to Figure 2. It’s not immediately clear why the AskAct reasoning is correct (the second highlighted think[] sentence on the right seems to just reiterate the query, similar to the first highlighted think[] sentence on the left). The item observations from the search don’t seem to indicate which option matches the query (they both are <40 dollars and have mn4 as an option).

---

> ### Author Response · Authors · 2023-11-20
> **[1/3] Response to Reviewer EvHL**
>
> We thank the reviewer for his/her comments and suggestions.
>
> **[Responses to Weaknesses]**
>
> **W1.** Further experiments disentangling this would be helpful: for example, if we used the same LLM critics and trajectory ranking process with the ReAct prompt, would it perform on par with Prospector (these experiments seem to be present for ALFWorld but not WebShop)?
>
> **WA1.** To address this important suggestion, we conducted **additional experiments on WebShop**. To provide a more clear view on the performance of each component of Prospector, we conducted experiments with four different settings: (1) ReAct only, (2) AskAct only, (3) ReAct + Trajectory Ranking (TR), and (4) AskAct + TR. To expedite experiments and reduce costs, we used open-source LLMs (i.e., Llama-2-70B for the LLM Actor and FLAN-T5-3B for the LLM Critic) instead of closed-source LLMs. The additional experiment results are summarized in the bottom section of the following table.
>
> Some findings from the additional experiments can be summarized as follows:
> - On the WebShop environment, **Llama-2-70B**, one of representative open-source LLMs **can achieve comparable performance with text-davinci-002**, one of the most powerful LLMs.
> - In both cases of text-davinci-002 and Llama-2-70B, **AskAct meaningfully improves the success rate compared to ReAct:** from 35.8 to 39.8 on text-davinci-002, and **from 37.6 to 42.2 on Llama-2-70B**. This means that AskAct, a simple prompting method that adds extra question prompts on ReAct, can be effective.
> - **ReAct + TR (the setting requested by the reviewer) can improve ReAct from 37.6 to 42.2** in the success rate. However, ReAct + TR is less efficient in terms of the computation complexity, since **it requires 8x larger computation than AskAct** to achieve the similar performance.
> - **AskAct + TR further improves the success rate of AskAct (from 42.2 to 43.6)**, and provides better performance than ReAct + TR (42.2).
>
> We will add the results in the revised paper, and upload it during the author response period.
>
> | Method | LLM Actor | LLM Critic | Reward | Success Rate |
> |---|---|---|:---:|:---:|
> | Human (expert) | - | - | 82.1 | 59.6 |
> | Human (average) | - | - | 75.5 | 50.0 |
> | IL | - | - | 59.9 | 29.1 |
> | IL + RL | - | - | 62.4 | 28.7 |
> | ReAct | PaLM-540B | - | 66.6 | 40.0 |
> | ReAct | text-davinci-002 | - | 63.3 | 35.8 |
> | ReAct + Reflexion (k=8) | text-davinci-002 | - | - | 35.0 |
> | AskAct (Prospector) | text-davinci-002 | - | 66.5 | 39.8 |
> | **AskAct + TR (k=8) (Prospector)** | text-davinci-002 | text-davinci-002 | 69.3 | **41.4** |
> | ReAct | Llama-2-70B | - | 62.3 | 37.6 |
> | AskAct (Prospector) | Llama-2-70B | - | 68.6 | 42.2 |
> | ReAct + TR (k=8) (Prospector) | Llama-2-70B | FLAN-T5-3B (SFT) | 69.3 | 42.2 |
> | **AskAct + TR (k=8) (Prospector)** | Llama-2-70B | FLAN-T5-3B (SFT) | 70.2 | **43.6** |

---

> ### Author Response · Authors · 2023-11-20
> **[2/3] Response to Reviewer EvHL**
>
> **[Responses to Weaknesses]**
>
> **W2.** In particular, it seems like AskAct was not applied to the ALFWorld benchmark for the Prospector agent, and only tested in WebShop as a single fixed question that asks "which observed object is most proper to select" (shown in Figure 2 and Table 13 and 16). Further experiments eliciting different types of self-asked questions across all the tasks would strengthen this contribution.
>
> **WA2.** To reflect this important suggestion, we conducted **additional experiments on ALFWorld**. Similar to the addtional experiments on WebShop, we conducted experiments with four different settings: (1) ReAct only, (2) AskAct only, (3) ReAct + Trajectory Ranking (TR), and (4) AskAct + TR.
>
> Some findings from the additional experiments can be summarized as follows:
> - **AskAct effectively improves the success rate of ReAct (from 41.0 to 56.7)**. Note that a LLM with lower temperature provides slightly better performance in case of single sampling.
> - Since AskAct provides a better baseline, **AskAct + TR can achieve much better performance with less sampling** (e.g., AskAct only (56.7) comparable with ReAct + TR (k=2) (56.0)).
> - We emphasize that **AskAct and TR can make an effective synergy in improving LLM agents in terms of both performance and efficiency**.
>
> We will add the results and the AskAct prompt for ALFWorld in the revised paper.
>
> | Method | LLM Actor | LLM Critic | k=1 | 2 | 3 | 4 | 5 |
> | --- | --- | --- | :---: | :---: | :---: | :---: | :---: |
> | BUTLER (RL) | -| - | 37.0 | - | - | - | - |
> | Act | PaLM-540B | - | 45.0 | - | - | - | - |
> | ReAct | PaLM-540B | - | 70.9 | - | - | - | - |
> | ReAct | text-davinci-002 (temp=0.0) | - | 78.4 | - | - | - | - |
> | ReAct + Reflexion | text-davinci-002 | - | - | - | - | - | 86.0 |
> | **ReAct + TR (Prospector)** | text-davinci-002 (temp=0.8) | text-davinci-002 | 71.6 | - | 93.3 | - | **95.5** |
> | ReAct | Llama-2-70B (temp=0.0) | - | **41.0** | - | - | - | - |
> | AskAct (Prospector) | Llama-2-70B (temp=0.0) | - | **56.7** | - | - | - | - |
> | ReAct + TR (Prospector) | Llama-2-70B (temp=0.8) | FLAN-T5-3B (SFT) | 33.6 | 59.0 | 69.4 | 73.1 | **77. 6** |
> | **AskAct + TR (Prospector)** | Llama-2-70B (temp=0.8) | FLAN-T5-3B (SFT) | 53.7 | 76.1 | 80.6 | 84.3 | **86.6** |

---

> > ### Comment · Reviewer_EvHL · 2023-11-22
> >
> > Thank you for addressing my concerns and presenting the additional experimental results, which are more thorough. Given these updated results, I have two remaining questions:
> > - Is there an explanation for why there exists a performance difference between ReAct/AskAct only and their respective cases with TR and k=1? I would expect those to be similar, but it seems like TR with k=1 actually hurts performance.
> > - Could the authors provide an example of the self-ask questions generated for ALFWorld?

---

> ### Author Response · Authors · 2023-11-20
> **[3/3] Response to Reviewer EvHL**
>
> **[Responses to Questions]**
>
> **Q1.** It would be helpful to have an ablation with the trajectory ranking only, and no intermediate self-asking step for WebShop, as is shown in ALFWorld. Is most of the juice coming the critic-based trajectory ranking?
>
> **A1.** This question is closely related to the weakness 1. Please see our response to the weakness 1 above.
>
> ---
> **Q2.** It would also be interesting to see if the few-shot critic performance improves with intermediate chain-of-thought reasoning steps in the critic prompt, rather than just having the critic immediately output a response as success/failure.
>
> **A2.** Thank you for your thoughtful suggestion. We will try our best to provide some experiment results on this suggestion within the author response period.
>
> ---
> **Q3.** It would help to add a clarifying caption to Figure 2.
>
> **A3.** Thank you for your constructive suggestion. We will add a clarifying caption to Figure 2 in the revised paper. We plan to upload the revised paper during the author response period.

---

> ### Author Response · Authors · 2023-11-22
> **Response to Reviewer EvHL**
>
> Thank you for your thoughtful comments. We provide answers to your remaining questions.
>
> ---
> **Q1**. Is there an explanation for why there exists a performance difference between ReAct/AskAct only and their respective cases with TR and k=1? I would expect those to be similar, but it seems like TR with k=1 actually hurts performance.
>
> **A1**. As you point out, there is a slight performance difference between ReAct (AskAct) only and ReAct (AskAct) + TR (k=1). It is due to that we set a different temperature of the LLM Critic (Llama-2-70B) for each case. **To leverage the stochastic generation of LLMs, Prospector sets a high temperate (0.8), when applying Trajectory Ranking (TR).** In contrast, ReAct sets the temperature of a LLM to 0.0 (i.e., deterministic sampling). Sampling from the LLM with a high temperature usually results in a slightly lower performance, since the probability distribution of tokens becomes to have higher uncertainty. Instead, we can generate diverse trajectories, and one of which can be highly likely correct. Actually, we cannot leverage the stochasticity of a LLM in case of TR with k=1. However, to make consistency, we set the same high temperature of a LLM when applying TR.
>
> ---
> **Q2**. Could the authors provide an example of the self-ask questions generated for ALFWorld?
>
> **A2**. Thank you for your request. We are happy to provide an example of AskAct prompt for ALFWorld. An AskAct prompt is based on a ReAct prompt, and it simply adds few self-asking steps before reasoning steps. In the following example AskAct prompt, we highlight the added self-asking steps. **In ALFWorld, the self-asking step checks if a target object is found.** This self-asking step significantly reduces missing a target object in a place (due to hallucination). Further, it can elicit a proper next action, if the target object is found. **We would like to emphasize that this simple prompting can significantly improves the success rate in ALFWorld (on Llama-2-70B, ReAct: 41.0, AskAct: 56.7)**. Here is an example AskAct prompt.
>
> ---
>
> You are in the middle of a room. Looking quickly around you, you see a cabinet 4, a cabinet 3, a cabinet 2, a cabinet 1, a countertop 1, a garbagecan 1, a handtowelholder 2, a handtowelholder 1, a sinkbasin 2, a sinkbasin 1, a toilet 1, a toiletpaperhanger 1, and a towelholder 1.
>
> Your task is to: put some spraybottle on toilet.
>
> \> think: To solve the task, I need to find and take a sparybottle, then put it on toilet.
>
> OK.
>
> \> think: First I need to find a spraybottle. A spraybottle is more likely to appear in cabinet (1-4), countertop (1), toilet (1), sinkbasin (1-2), garbagecan (1). I can check one by one, starting with cabinet 1.
>
> OK.
>
> \> go to cabinet 1
>
> On the cabinet 1, you see a cloth 1, a soapbar 1, a soapbottle 1.
>
> \> **think: Did I find a spraybottle?**
>
> OK.
>
> \> **think: No, I did not. I need to go other place to find a spraybottle.**
>
> OK.
>
> \> go to cabinet 2
>
> The cabinet 2 is closed.
>
> \> open cabinet 2
>
> You open the cabinet 2. The cabinet 2 is open. In it, you see a candle 1, and a spraybottle 2.
>
> \> **think: Did I find a spraybottle?**
>
> OK.
>
> \> think: Now I find a spraybottle (2). Next, I need to take it.
>
> OK.
>
> \> take spraybottle 2 from cabinet 2
>
> You pick up the spraybottle 2 from the cabinet 2.
>
> \> think: Now I take a spraybottle (2). Next, I need to put it in/on toilet 1.
>
> OK.
>
> \> go to toilet 1
>
> On the toilet 1, you see a soapbottle 2.
>
> \> put spraybottle 2 in/on toilet 1
>
> You put the spraybottle 2 in/on the toilet 1.

---

### Official Review · Reviewer_EsKc · 2023-10-31

**Soundness:** 2 fair
**Presentation:** 3 good
**Contribution:** 2 fair
**Rating:** 5
**Confidence:** 3

**Summary:**

The paper introduces Prospector, an innovative LLM agent designed for decision-making tasks. Unlike previous methods such as ReAct and Reflexion, which rely on few-shot in-context learning or use feedback from the environment, Prospector integrates two novel components: Self-Asking and Trajectory Ranking. Self-Asking allows the LLM to pose and answer its own questions during few-shot demonstrations, aiming to collect more pertinent information for decision-making. Trajectory Ranking, on the other hand, involves generating multiple action trajectories and selecting the most rewarding one using reward prediction models. The authors show that Prospector significantly outperforms existing methods on benchmark tasks like ALFWorld and WebShop.

**Strengths:**

1. The paper addresses a gap in current LLM-based decision-making methods by integrating feedback from the environment and incorporating stochasticity in trajectory generation.
2. The proposed method shows empirical success, outperforming existing state-of-the-art methods on standard benchmarks.
3. Prospector offers an approach that avoids costly fine-tuning, making it more generalizable and efficient.

**Weaknesses:**

1. Both the critic and the generator are LLMs. This could amplify any existing issues inherent to LLMs.
2. Limited discussion on the limitations of the reward prediction models used for Trajectory Ranking.
3. The paper could benefit from a more comprehensive analysis comparing the computational overhead introduced by the Self-Asking and Trajectory Ranking components.

**Questions:**

1. How does the computational complexity of Prospector compare to that of existing methods like ReAct and Reflexion?
2. Could you elaborate on the reward prediction models used in Trajectory Ranking? What are the limitations of the reward prediction models you used, and how do they impact the overall performance of Prospector?
3. Are there specific types of questions or domains where the Self-Asking mechanism is more or less effective?

---

> ### Author Response · Authors · 2023-11-22
> **Response to Reviewer EsKc**
>
> We thank the reviewer for his/her comments and suggestions.
>
> **[Responses to Questions]**
>
> **Q1.** How does the computational complexity of Prospector compare to that of existing methods like ReAct and Reflexion?
>
> **A1.** We present an analysis of the complexity of each method.
> Let $N$ be the number of tokens involved in a single trajectory and $m$ the number of generated trajectories. Assuming a transformer model is used as the backbone, the complexity of ReAct is $O(N^2)$.
> Reflexion generates a sequence of $m$ trajectories. Assuming all past trajectories are given to the model, Reflexion has a complexity of $O(m^2N^2)$. Note that this may vary depending on how the method is implemented.
> Our method involves sampling $m$ independent trajectories and ranking them. This process has a complexity of $O(mN^2)$. **Although ReAct is more efficient than our approach, our method performs much better.** **Our method is further more efficient and performs better than Reflexion based on the analysis above.**
>
> ---
>
> **Q2.** Could you elaborate on the reward prediction models used in Trajectory Ranking? What are the limitations of the reward prediction models you used, and how do they impact the overall performance of Prospector?
>
> **A2.** We use a LLM as a reward model (named LLM Critic) that predicts the expected reward of a trajectory generated by the LLM Actor. **In this paper, we consider two types of LLM Critics: (1) few-shot LLM Critic and (2) fine-tuned LLM Critic.** A few-shot LLM Critic takes few examples of (trajectory, reward) as inputs, and generates a response as a reward. A fine-tuned LLM Critic is fine-tuned on a dataset of (trajectory, reward), and generate response as the reward for a given input trajectory. A few-shot LLM Critic has some advantages of generality, since it does not need to be trained on a specific environment. But, it may provide low prediction accuracy, as shown in the case of the WebShop environment. In contrast, a fine-tuned LLM Critic can provide high reward prediction accuracy, since it is trained on trajectories generated from an environment. Recently, building a general reward model is investigated as an important research problem. One of the representative work is Prometheus [1]. **We believe that Prospector can benefit from the general and accurate reward models like Prometheus.**
>
> [1] Seungone Kim, et al., "Prometheus: Inducing Fine-grained Evaluation Capability in Language Models", arXiv:2310.08491.
>
> ---
>
> **Q3.** Are there specific types of questions or domains where the Self-Asking mechanism is more or less effective?
>
> **A3.** In this study, **we find that Self-Asking is considerably effective in both ALFWorld and WebShop.** To answer this thoughtful question, we conducted additional experiments on ALFWorld. As shown in the following table, **AskAct effectively improves the success rate of ReAct (from 41.0 to 56.7).**
>
> | Method | LLM Actor | LLM Critic | Success Rate (%) |
> | --- | --- | --- | --- |
> | ReAct | Llama-2-70B | - | **41.0** |
> | ReAct + TR (k=5) (Prospector) | Llama-2-70B | FLAN-T5-3B (SFT) | 77.6 |
> | AskAct | Llama-2-70B | - | **56.7** |
> | AskAct + TR (k=5) (Prospector) | Llama-2-70B | FLAN-T5-3B (SFT) | 86.6 |
>
> Self-Asking is a prompting method based on a ReAct prompt, and **it simply adds few self-asking steps before reasoning steps**. In the following example AskAct prompt for ALFWorld, we highlight the added self-asking steps. **In ALFWorld, the self-asking step checks if a target object is found.** This self-asking step significantly reduces missing a target object in a place (due to hallucination). Further, it can elicit a proper next action, if the target object is found.
>
> ---
>
> You are in the middle of a room. Looking quickly around you, you see a cabinet 4, a cabinet 3, a cabinet 2, a cabinet 1, a countertop 1, a garbagecan 1, a handtowelholder 2, a handtowelholder 1, a sinkbasin 2, a sinkbasin 1, a toilet 1, a toiletpaperhanger 1, and a towelholder 1.
>
> Your task is to: put some spraybottle on toilet.
>
> \> think: To solve the task, I need to find and take a sparybottle, then put it on toilet.
>
> OK.
>
> \> think: First I need to find a spraybottle. A spraybottle is more likely to appear in cabinet (1-4), countertop (1), toilet (1), sinkbasin (1-2), garbagecan (1). I can check one by one, starting with cabinet 1.
>
> OK.
>
> \> go to cabinet 1
>
> On the cabinet 1, you see a cloth 1, a soapbar 1, a soapbottle 1.
>
> \> **think: Did I find a spraybottle?**
>
> OK.
>
> \> **think: No, I did not. I need to go other place to find a spraybottle.**
>
> OK.
>
> \> go to cabinet 2
>
> The cabinet 2 is closed.
>
> \> open cabinet 2
>
> You open the cabinet 2. The cabinet 2 is open. In it, you see a candle 1, and a spraybottle 2.
>
> \> **think: Did I find a spraybottle?**
>
> OK.
>
> \> **think: Now I find a spraybottle (2). Next, I need to take it.**
>
> OK.
>
> \> take spraybottle 2 from cabinet 2

---

> > ### Comment · Reviewer_EsKc · 2023-11-22
> >
> > I still have concerns about the complexities and technical novelties. I will maintain my current scores for now.

---

### Official Review · Reviewer_UFEP · 2023-10-31

**Soundness:** 2 fair
**Presentation:** 2 fair
**Contribution:** 1 poor
**Rating:** 5
**Confidence:** 2

**Summary:**

In this paper, the authors present Prospector, an algorithm for improving LLM agents operating on language tasks. The authors introduce the concepts of Self-Asking, involving asking the model to ponder the task and options at hand, and trajectory ranking, which considers multiple rollouts from the language models and compares them using a "critic", which is a different LLM call, to find the best possible response.

The paper is organized as following: first, the interactive decision making tasks is introduced on which the rest of the paper is based on. Then, the authors introduce the two new components in this paper, namely self-asking, where the agent asks and answers a question about the task at hand, and trajectory ranking, where multiple natural language interactions are rolled out, and then the best trajectory is picked out by an LLM critic. The authors present a variety of experiments based on these premises, such as how different language models perform as the action model, and how well different critics (fine tuned vs. few shot prompted) perform against each other. The authors show that while it may not be very important to fine tune the actor, fine tuning the critic leads to clear improvements. The necessary ablations (such as: how well does each of these components perform by themselves?) are not marked separately, but included in the primary reported results tables.

**Strengths:**

The paper has a few strong points, such as:
1. A comprehensive evaluation across different language models used as critics.
2. On different parameters of the experiments, a proper experimentation schedule was used, such as few-shot reward prediction accuracy.
3. The success rate on the evaluated benchmarks show marked improvement over previous work, however, I am not familiar with the benchmarks in the field enough to know if this is sufficient.

**Weaknesses:**

The positive impact of the paper is beset by several downsides. Here are these in the order of importance:
1. I am not certain about the magnitude of the impact of the method introduced in this paper. The method of self-asking itself does not seem significant enough in and of itself without the trajectory ranking, and is quite similar to many different previous methods such as thinking step by step. Trajectory ranking is definitely the more interesting of the two components, but I am not sure it is a novel and significant enough contribution to merit a place in this venue.
2. Following up on this, the work is beset by the fact that the new methods are only evaluated in two benchmarks only. While they perform well on the benchmarks, the question of how easy they will be to scale to a variety of other tasks remain unanswered from the paper itself.
3. While there is a comprehensive study run on LLM critic and which language model is best for that task, it does not extend to the LLM actor itself. Rather, only two models of incredibly large sizes are used, which keeps the evaluation quite one-sided.

Overall, this paper shows promise in a few direction, but does not make a noteworthy contribution in any of the directions in my opinion. However, given my limited experience in such works, I am happy to reconsider my take at the word of the area chair.

**Questions:**

1. How well does the open source smaller models perform on the tasks presented in the paper?
2. Could you please expand the previous work section to properly differentiate yourself from them and clarify what contributions of yours in this paper are novel vs. same as what is done before?
3, What would be the primary challenges of scaling this method to new benchmarks?

---

> ### Author Response · Authors · 2023-11-20
> **Response to Reviewer UFEP**
>
> We thank the reviewer for his/her comments and suggestions.
>
> **[Responses to Questions]**
>
> **Q1.** How well does the open source smaller models perform on the tasks presented in the paper?
>
> A1. To answer this important question, we conducted **comprehensive additional experiments on ALFWorld and WebShop by using open-source LLMs**. More specifically, we used **Llama-2-70B for the LLM Actor** and **fined-tuned FLAN-T5-3B for the LLM Critic**. We would like to emphasize that AskAct and TR can make an effective synergy in improving LLM agents in terms of both performance and efficiency.
>
> The **open-source LLM experiments on ALFWorld** can be summarized as follows:
> - **AskAct effectively improves the success rate of ReAct (from 41.0 to 56.7)**.
> - Since AskAct provides a better baseline, **AskAct + TR can achieve much better performance with less sampling** (e.g., AskAct only (56.7) comparable with ReAct + TR (k=2) (56.0)).
>
> | Method | LLM Actor | LLM Critic | k=1 | 2 | 3 | 4 | 5 |
> | --- | --- | --- | :---: | :---: | :---: | :---: | :---: |
> | ReAct | text-davinci-002 (temp=0.0) | - | 78.4 | - | - | - | - |
> | ReAct + Reflexion | text-davinci-002 | - | - | - | - | - | 86.0 |
> | **ReAct + TR (Prospector)** | text-davinci-002 (temp=0.8) | text-davinci-002 | 71.6 | - | 93.3 | - | **95.5** |
> | ReAct | Llama-2-70B (temp=0.0) | - | **41.0** | - | - | - | - |
> | AskAct (Prospector) | Llama-2-70B (temp=0.0) | - | **56.7** | - | - | - | - |
> | ReAct + TR (Prospector) | Llama-2-70B (temp=0.8) | FLAN-T5-3B (SFT) | 33.6 | 59.0 | 69.4 | 73.1 | **77. 6** |
> | **AskAct + TR (Prospector)** | Llama-2-70B (temp=0.8) | FLAN-T5-3B (SFT) | 53.7 | 76.1 | 80.6 | 84.3 | **86.6** |
>
> ---
> The **open-source LLM experiments on WebShop** can be summarized as follows:
> - On the WebShop environment, **Llama-2-70B**, one of representative open-source LLMs **can achieve comparable performance with text-davinci-002**, one of the most powerful LLMs.
> - In both cases of text-davinci-002 and Llama-2-70B, **AskAct meaningfully improves the success rate compared to ReAct:** from 35.8 to 39.8 on text-davinci-002, and **from 37.6 to 42.2 on Llama-2-70B**. This means that AskAct, a simple prompting method that adds extra question prompts on ReAct, can be effective.
> - **AskAct + TR further improves the success rate of AskAct (from 42.2 to 43.6)**, and provides better performance than ReAct + TR (42.2).
>
> | Method | LLM Actor | LLM Critic | Reward | Success Rate |
> |---|---|---|:---:|:---:|
> | ReAct | text-davinci-002 | - | 63.3 | 35.8 |
> | ReAct + Reflexion (k=8) | text-davinci-002 | - | - | 35.0 |
> | AskAct (Prospector) | text-davinci-002 | - | 66.5 | 39.8 |
> | **AskAct + TR (k=8) (Prospector)** | text-davinci-002 | text-davinci-002 | 69.3 | **41.4** |
> | ReAct | Llama-2-70B | - | 62.3 | 37.6 |
> | AskAct (Prospector) | Llama-2-70B | - | 68.6 | 42.2 |
> | ReAct + TR (k=8) (Prospector) | Llama-2-70B | FLAN-T5-3B (SFT) | 69.3 | 42.2 |
> | **AskAct + TR (k=8) (Prospector)** | Llama-2-70B | FLAN-T5-3B (SFT) | 70.2 | **43.6** |
>
> We will add the results in the revised paper, and upload it during the author response period.

---

### Official Review · Reviewer_TbNU · 2023-11-04

**Soundness:** 2 fair
**Presentation:** 3 good
**Contribution:** 2 fair
**Rating:** 6
**Confidence:** 3

**Summary:**

The authors propose Prospector, which extends in-context learning (ICL) for LLMs to be able to optimize trajectories based on reward from environment using self-asking and ranking in decision making tasks. On two decision-making benchmark environments such as ALFWorld and WebShop, they empirically demonstrate that Prospector can considerably increase the success rate of given tasks, while outperforming recent advancements such as ReAct and Reflexion.

**Strengths:**

1. The paper is well-written and easy to follow
2. Extensive ablations and analysis presented is nicely done -- it shows how the two components of the prospector framework work and improve the performance of the baseline react models.

**Weaknesses:**

1. Limited novelty: While it is good to see how two simple ideas when put together in the prospector framework can lead to good task performance in interactive decision making scenarios, the two ideas themselves are very close to existing work. Consequently, the novelty seems a bit limited, IMO.
2. Broader baselines: I liked the authors ablations and comparison with React and its variants given the closeness of the approach (prospector) to react.  These were helpful in understanding how prospector's individual components improve performance. However, it would have been also useful to see how prospector's performance compares to other llm planning approaches e.g., the ones that combine llms + tree search/classical planning approaches such as https://arxiv.org/pdf/2307.08962.pdf to see how far does prospector push the performance. Lastly, given that prospector does some training for critics using example trajectories, I am wondering how the performance of prospector would compare to finetuned LLM planner/policy e.g., with LIMA (https://arxiv.org/abs/2305.11206) which can be used to finetune LLM with limited data. Without these, right now, it is unclear whether prospector should be the goto planning approach for interactive decision making problems or is it really just a better version of react?

**Questions:**

- Unclear why the authors do not show prospector with askact + trajectory ranking results on Alfworld (Table 2,4). I’d encourage  the authors to do this for the sake of completeness. Likewise, it would be good to see prospector with react + trajectory ranking on webshop (table 7).
- Table 5,8: Why is few-shot reward prediction accuracy of LLM critic lower with more shots (3-shot vs. 2-shot)?
- It seems that Prospector would be slower than React or reflexion because of additional reasoning that it does using more LLM calls. For real world interactive decision making tasks, it might be useful for the authors to also report compute time needed to decide the next action during the task execution. To that end, it would be great to also add a limitation section.
- What is the advantage of the LLM critic over a “learnt” critic which can take a policy rollout and provide a corresponding reward? Given that prospector is evaluated only in sim environment, why not use sim to learn such a critic?
- The authors dont seem to cite or mention self-refine: https://arxiv.org/pdf/2303.17651.pdf but that seemed very similar to self-asking in prospector too IMO.
- Opensourcing plans? Despite the simplicity of the approach, I encourage the authors to opensource their code for reproducibility.

---

> ### Author Response · Authors · 2023-11-22
> **Response to Reviewer TbNU**
>
> We thank the reviewer for his/her comments and suggestions.
>
> **[Responses to Questions]**
>
> **Q1.** Unclear why the authors do not show prospector with askact + trajectory ranking results on Alfworld (Table 2,4).
>
> **A1.** To answer this important question, we conducted **additional experiments on ALFWorld**. We conducted experiments with four different settings: (1) ReAct only, (2) AskAct only, (3) ReAct + Trajectory Ranking (TR), and (4) AskAct + TR. To expedite experiments and reduce costs, we used open-source LLMs (i.e., Llama-2-70B for the LLM Actor and FLAN-T5-3B for the LLM Critic) instead of closed-source LLMs. The additional experiment results are summarized in the bottom section of the following table.
>
> Some findings from the additional experiments can be summarized as follows:
> - **AskAct effectively improves the success rate of ReAct (from 41.0 to 56.7)**. Note that a LLM with lower temperature provides slightly better performance in case of single sampling.
> - **Since AskAct provides a better baseline, AskAct + TR can achieve much better performance with less sampling** (e.g., AskAct only (56.7) comparable with ReAct + TR (k=2) (56.0)).
> - We emphasize that **AskAct and TR can make an effective synergy in improving LLM agents in terms of both performance and efficiency**.
>
> | Method | LLM Actor | LLM Critic | k=1 | 2 | 3 | 4 | 5 |
> | --- | --- | --- | :---: | :---: | :---: | :---: | :---: |
> | ReAct | text-davinci-002 (temp=0.0) | - | 78.4 | - | - | - | - |
> | ReAct + Reflexion | text-davinci-002 | - | - | - | - | - | 86.0 |
> | **ReAct + TR (Prospector)** | text-davinci-002 (temp=0.8) | text-davinci-002 | 71.6 | - | 93.3 | - | **95.5** |
> | ReAct | Llama-2-70B (temp=0.0) | - | **41.0** | - | - | - | - |
> | AskAct (Prospector) | Llama-2-70B (temp=0.0) | - | **56.7** | - | - | - | - |
> | ReAct + TR (Prospector) | Llama-2-70B (temp=0.8) | FLAN-T5-3B (SFT) | 33.6 | 59.0 | 69.4 | 73.1 | **77. 6** |
> | **AskAct + TR (Prospector)** | Llama-2-70B (temp=0.8) | FLAN-T5-3B (SFT) | 53.7 | 76.1 | 80.6 | 84.3 | **86.6** |
>
> ---
>
> **Q2.** Table 5,8: Why is few-shot reward prediction accuracy of LLM critic lower with more shots (3-shot vs. 2-shot)?
>
> **A2.** This is due to the **limited context length** of text-davinci-002 (i.e., 4096). In both ALFWorld and WebShop, 3-shot examples often exceed the context length, and the last example is truncated.
>
> ---
>
> **Q3.** It seems that Prospector would be slower than React or reflexion because of additional reasoning that it does using more LLM calls.
>
> **A3.** Let $N$ be the number of tokens involved in a single trajectory and $m$ the number of generated trajectories. Assuming a transformer model is used as the backbone, the complexity of ReAct is $O(N^2)$. Reflexion generates a sequence of $m$ trajectories. Assuming all past trajectories are given to the model, Reflexion has a complexity of $O(m^2N^2)$. Note that this may vary depending on how the method is implemented. Our method involves sampling $m$ independent trajectories and ranking them. This process has a complexity of $O(mN^2)$. **Although ReAct is more efficient than our approach, our method performs much better.** **Our method is further more efficient and performs better than Reflexion based on the analysis above.**
>
> ---
>
> **Q4.** For real world interactive decision making tasks, it might be useful for the authors to also report compute time needed to decide the next action during the task execution. To that end, it would be great to also add a limitation section.
>
> **A4.** Thank you for your thoughtful question. **We believe that AskAct prompting (Self-Asking) can alleviate the computation overhead of Trajectory Ranking.** Since AskAct prompting provides considerably better baseline than ReAct prompting, AskAct + TR can achieve reasonable performance with less sampling. We can see this effect in the experiment results in the above table.
>
> ---
>
> **Q5.** What is the advantage of the LLM critic over a “learnt” critic which can take a policy rollout and provide a corresponding reward? Given that prospector is evaluated only in sim environment, why not use sim to learn such a critic?
>
> **A5.** We would like to point out that this is one of the critics we consider in our paper. As we discuss in the last paragraph of section 3.2, **the in-context learning based critic fails to perform well in complex environments and a fine-tuned critic model performs better in this scenario.**
>
> ---
>
> **Q6.** The authors don't seem to cite or mention self-refine.
>
> **A6.** Thank you for pointing out this related work, we will include a discussion about the Self-Refine paper in our revision.
>
> ---
>
> **Q7.** Opensourcing plans?
>
> **A7.** We plan to opensource our implementation upon paper acceptance.

---

> ### Comment · Reviewer_TbNU · 2023-11-22
> **Post rebuttal**
>
> Thanks for the additional ablations and comparisons.
>
> While I understand that prospector is more efficient than reflexion, I am still not clear about the computation time/number of LLM calls needed (on average) before an action is obtained as output from Prospector. This will be important for real-world deployment.
>
> I'd also like the authors to comment on the novelty of their contributions since I see that other reviewers have also raised similar concerns.
>
> Given the above, I'll hold my score for now. I think the paper is interesting but I cant champion the paper.

---

### Meta-Review · Area_Chair_cfSV · 2023-12-03

**Metareview:**

The paper received borderline ratings from the reviewers. The reviewers raised several concerns such as:
- Limited novelty
- Lack of clarity on the number of LLM calls needed to output an action
- Critic and generator both being LLMs
- Limited discussion on the limitations of the reward prediction models
- Applying AskAct to only one of the benchmarks

The rebuttal addressed some of these issues and provided additional experiments that were helpful. However, the rebuttal falls short in terms of addressing the novelty issue, a concern emphasized by almost all reviewers. Additionally, there are still missing details about the complexity and efficiency of the method (as mentioned in the reviewers’ responses to the rebuttal). These issues preclude acceptance, and the AC recommends rejection.

**Justification For Why Not Higher Score:**

While the paper and the rebuttal showcase good performance, they fall short in adequately justifying the novelty of the contributions in comparison to prior work.

**Justification For Why Not Lower Score:**

N/A

---

### Decision · Program_Chairs · 2024-01-16

Reject